# Establishment of Polydopamine-Modified HK-2 Cell Membrane Chromatography and Screening of Active Components from *Plantago asiatica* L.

**DOI:** 10.3390/ijms25021153

**Published:** 2024-01-18

**Authors:** Hongxue Gao, Zhiqiang Liu, Fengrui Song, Junpeng Xing, Zhong Zheng, Zong Hou, Shu Liu

**Affiliations:** 1State Key Laboratory of Electroanalytical Chemistry & Jilin Provincial Key Laboratory of Chinese Medicine Chemistry, Changchun Institute of Applied Chemistry, Chinese Academy of Sciences, Changchun 130022, China; mslab34@ciac.ac.cn (H.G.);; 2Institute of Applied Chemistry and Engineering, University of Science and Technology of China, Hefei 230029, China

**Keywords:** cell membrane chromatography, silica gel, polydopamine, *Plantago asiatica* L., uric acid

## Abstract

Cell membrane chromatography (CMC) has been widely recognized as a highly efficient technique for in vitro screening of active compounds. Nevertheless, conventional CMC approaches suffer from a restricted repertoire of cell membrane proteins, making them susceptible to oversaturation. Moreover, the binding mechanism between silica gel and proteins primarily relies on intermolecular hydrogen bonding, which is inherently unstable and somewhat hampers the advancement of CMC. Consequently, this investigation aimed to establish a novel CMC column that could augment protein loading, enhance detection throughput, and bolster binding affinity through the introduction of covalent bonding with proteins. This study utilizes polydopamine (PDA)-coated silica gel, which is formed through the self-polymerization of dopamine (DA), as the carrier for the CMC column filler. The objective is to construct the HK-2/SiO_2_-PDA/CMC model to screen potential therapeutic drugs for gout. To compare the quantity and characteristics of Human Kidney-2 (HK-2) cell membrane proteins immobilized on SiO_2_-PDA and silica gel, the proteins were immobilized on both surfaces. The results indicate that SiO_2_-PDA has a notably greater affinity for membrane proteins compared to silica gel, resulting in a significant improvement in detection efficiency. Furthermore, a screening method utilizing HK-2/SiO_2_-PDA/CMC was utilized to identify seven potential anti-gout compounds derived from *Plantago asiatica* L. (PAL). The effectiveness of these compounds was further validated using an in vitro cell model of uric acid (UA) reabsorption. In conclusion, this study successfully developed and implemented a novel CMC filler, which has practical implications in the field.

## 1. Introduction

The technique of Cell membrane chromatography (CMC) was initially introduced by He et al. [1,2]. CMC is a biomimetic affinity chromatography method that employs active cell membrane receptors as the stationary phase, simulating the interactions between drugs and cell membrane receptors in vitro [3,4]. The combination of CMC and mass spectrometry (MS) provides comprehensive advantages in complex compound separation and active compound screening, making it a convenient, fast, stable, and highly sensitive method. This technique is particularly well-suited for the identification of active compounds within intricate systems such as traditional Chinese medicine (TCM) [5,6].

The synthesis of cell membrane stationary phase (CMSP) materials involves combining the cell membrane with the silicone hydroxyl groups (Si-OH) present on the surface of silica gel. This combination enables the maintenance of the cell membrane’s integrity, its three-dimensional structure, and the biological activity of its receptors [4,7]. In recent years, there has been a discernible rise in the development of novel CMSP materials with the specific objective of screening active constituents in traditional Chinese medicine. Particularly, cell membrane chromatography has made substantial progress in evaluating the efficacy of TCM by improving protein carrier and enhancing the binding of proteins with carriers [8,9,10]. Consequently, there has been a significant surge in research endeavors focused on the interaction between silica gel carriers and proteins, leading to the emergence of various innovative techniques that facilitate this process. Despite the notable progress made in CMC, there are still persistent challenges that hinder its further development. One limitation concerns the volume constraints inherent in cell membrane chromatography, which limit the number of proteins bound to silica gel. This limitation arises from the necessity of packing the chromatographic column with protein-bound silica gel, thereby restricting the size of the column due to inadequate cell quantity. This may give rise to the occurrence of oversaturation in the process of detection, impeding the examination of samples with high concentrations. This is because the presence of high concentration compounds may hinder the effective binding of trace compounds, posing challenges to the screening and detection of these compounds. In addition, the interaction between silica gel and protein mainly occurs through intermolecular hydrogen bonding, resulting in a relatively weak binding between protein and silica gel, thus making the protein prone to detachment from the silica gel. However, current research has rarely focused on improving the protein loading of cell membrane chromatography and enhancing the binding of silica gel to proteins. The objective of this study was to develop a novel protein-silica gel binding method and enhance the protein loading capacity of silica gel through the establishment of a new CMC model. Polydopamine (PDA) was chosen as the modifier for silica gel to improve detection throughput in this study. Previous research has demonstrated that dopamine can undergo self-polymerization under weak alkaline conditions, resulting in the formation of polydopamine on the surface of different materials, and has exhibited distinctive properties [11,12,13,14]. The simplicity of PDA synthesis and the presence of abundant active functional groups on its surface led to its selection as the modifier for silica gel, thereby establishing a novel CMC model by using silica gel modified with PDA.

Gout, a prevalent joint disease, is closely linked to the concentration of uric acid (UA) within the body [15,16]. The quantity of UA present in the body is typically regulated by the synthesis and elimination of UA in the renal system [17]. Within the human kidneys, the excretion of UA encompasses glomerular filtration, reabsorption, and proximal tubule secretion. Consequently, the level of UA within the human body is intimately associated with the reabsorption process of UA occurring within the kidneys [18]. In individuals with elevated levels of uric acid (UA), a significant proportion (90%) is attributed to renal mechanisms [19], specifically involving organic anion transporters (OATs) in the kidney, such as human urate anion exchanger 1 (URAT1) and human organic anion transporter 1 (hOAT1). Inhibition of these anion transporters may hinder UA reabsorption, resulting in a decrease in UA levels within the body and potentially alleviating or treating gout. Consequently, this experiment focuses on the crucial role of URAT1 in gout and utilizes the CMC method to identify potential anti-gout compounds from PAL.

This study utilized a PDA-based approach to modify silica gel and construct a CMC. Initially, the optimal ratio of PDA to silica gel in the newly formed CMSP was determined, followed by the verification and characterization of the binding between silica gel and membrane proteins through scanning electron microscope (SEM) and Fourier transform infrared spectrometer (FT-IR) analyses. Subsequently, positive and negative drugs were utilized to assess the effectiveness of SiO_2_-PDA/CMC. Additionally, the novel established CMC was used to screen potential anti-gout compounds from the P2 group of PAL. The P2 group of PAL was obtained via filtration and purification utilizing macroporous resin column chromatography, as detailed in the previous literature [20]. The potential active constituents were subsequently verified by evaluating the UA reabsorption of HK-2. And surface plasmon resonance (SPR) and molecular docking were used to verify the combination of acteoside and URAT1. The experimental modification of the CMC column technique is uncomplicated, resulting in a notable augmentation in the silica gel carrier protein content and detection throughput, as well as an improvement in the binding interaction between silica gel and protein. This advancement bears significant significance for the progression of the field.

## 2. Results and Discussion

### 2.1. Optimization of Combination Ratio of PDA and Silica Gel

Under weakly alkaline conditions, DA exhibits a unique capacity for self-polymerization on the surface of silica gel, thereby altering the surface properties and functional groups of silica gel and forming SiO_2_-PDA materials. It is necessary to detect the optimal ratio of DA and silica gel in order to enhance the detection throughput. During the experiment, various SiO_2_-PDA materials were produced by adjusting the ratio of DA to silica gel. Subsequently, the optimal ratio of DA to silica gel for maximum protein binding was identified.

Different SiO_2_-PDA materials were prepared by adding DA of different weight to 50 mg silica gel. Then, excessive cell membrane proteins were added to 40 mg of different SiO_2_-PDA materials to prepare CMSP, and the membrane proteins bound to SiO_2_-PDA were quantified by BCA kit (Figure 1). It was observed that the counts of cell membrane proteins bound to SiO_2_-PDA exhibited a gradual increase with increasing DA content, until reaching a maximum when the weight of DA reached 42 mg. When the weight of DA is 56 mg, there is a decrease in the binding of membrane proteins. The decrease in binding may be attributed to excessive accumulation of PDA, which covers the active site and reduces the active site per unit area. It is also possible that after combining with PDA, the diameter of silica gel increases, reducing the specific surface area of the silica gel and decreasing the maximum protein content that the silica gel can bind to. Consequently, the binding of SiO_2_-PDA to the cell membrane is reduced, resulting in a decrease in the efficiency of SiO_2_-PDA binding to cell membrane proteins. Therefore, the optimal binding ratio is achieved by combining 42 mg of DA with 50 mg of silica gel.

### 2.2. Optimization of Cell Dosage

The quantity of cells utilized during the preparation of a CMC column has an important impact on its efficiency and success rate. When an excessive number of cells are employed, the yield of extracted cell membranes increases under identical conditions, thereby exacerbating the inefficiency of the experimental procedure. On the contrary, insufficient cell count will lead to insufficient binding proteins on the surface of silica gel, resulting in an uneven distribution of silica gel and a decrease in column efficiency. Therefore, it is necessary to conduct a screening process to determine the suitable quantities of cells for usage. The purpose of this study was to determine the optimal cell quantity used in the preparation of CMC column, aiming to minimize cell dosage and preserving the effectiveness of the CMC column. This approach aids in conserving experimental resources and reducing costs, while simultaneously mitigating the risk of post-column plugging and ensuring optimal column performance.

The data presented in Figure 2 demonstrates that the quantity of cell membrane proteins bound to 40 mg silica gel or SiO_2_-PDA is subject to variation based on the number of cells. Figure 2A illustrates that once the cell count reaches 3.5 × 10^7^, the saturation point is reached for silica gel-bound membrane protein, resulting in no further increase in silica gel-bound proteins as the cell number increases. Conversely, Figure 2B reveals that 40 mg SiO_2_-PDA exhibits a higher protein binding capacity, reaching saturation only when the cell count reaches 16 × 10^7^. The phenomenon can potentially be attributed to the substantial accumulation of PDA on the silica gel surface, resulting in a notable augmentation of active sites and consequently enabling an increased binding capacity for membrane proteins. The existing literature on CMC predominantly focuses on enhancing the binding methods between carriers and proteins, while overlooking the optimization of the binding quantity. Compared with the previously reported findings in the literature [21,22], the number of proteins bound by silica gel in the experiment is close to that reported in the literature, and the content of proteins bound by SiO_2_-PDA is significantly increased.

### 2.3. Characterization of CMSP

Based on the observations presented in Figure 3A,B, it is evident that both silica gel and SiO_2_-PDA exhibit regular spherical shapes with smooth surfaces, and no significant changes are observed. However, upon binding to the membrane protein, protein adhesion is observed on the surfaces of both silica gel and PDA, indicating a strong binding affinity between the membrane protein and silica gel (as depicted in Figure 3C,D). Notably, the diameter of the silica gel in Figure 3C,D is found to be reduced compared to that in Figure 3A,B. Consequently, further investigation was undertaken to explore this phenomenon.

The SEM images are shown in Appendix A, where Appendix A exhibits silica gel, Appendix A illustrates silica gel after 5 min of vacuum stirring, and Appendix A displays silica gel after 60 min of suction filtration. The diameter of the silica gel is listed in Appendix A. It is evident that, compared to the silica gel depicted in Appendix A, the size of the silica gel in Appendix A decreases. This phenomenon can be attributed to the contraction of the internal pores of the silica gel after suction filtration, leading to a reduction in the diameter of the silica gel.

Silica gel and SiO_2_-PDA were characterized by Fourier transform infrared (FT-IR) spectrometer, as shown in Figure 3E. The observed peaks at 477 and 807 cm^−1^ were attributed to Si-O stretches, while the peak at 1102 cm^−1^ was attributed to O-Si-O stretches. Additionally, the peak at 3420 cm^−1^ was attributed to -OH stretches. Furthermore, the absorption bonds at 3050 and 3150 cm^−1^ were the characteristic peaks of the indole structure, which was contributed by PDA. The peak at 1420 cm^−1^ was attributed to C=C stretches on the benzene ring, and the absorption bond at 702 cm^−1^ was attributed to -CH stretches. The FT-IR data provides confirmation of the successful aggregation of PDA onto the surface of silica gel.

The pore size of silica gel and SiO_2_-PDA is 7.07 and 9.26 nm (Figure 3F). It can be seen that silica gel and SiO_2_-PDA are mesoporous materials (Figure 3G) and the BET surface area is 192.5 and 151.5 m^2^/g.

### 2.4. Comparison of Stability between Silica Gel and SiO_2_-PDA

To compare the stability of silica gel and SiO_2_-PDA binding to proteins, an ultrasonic cell crusher and BCA kit were used to detect the content of protein shedding in the solution, and the stability of silica gel and SiO_2_-PDA binding to proteins was compared. The conditions for ultrasonic cell crusher were set as follows: a power of 400 W, ultrasound applied for 1 s, with an interval of 19 s. After different ultrasound frequencies for silica gel and SiO_2_-PDA, it can be observed that after performing ultrasound 25 times on SiO_2_/CMSP (Figure 4), the proteins bound to the silica gel were almost completely detached. After 41 rounds of ultrasonic treatment, the membrane proteins bound to SiO_2_-PDA were almost completely detached. This is because SiO_2_-PDA forms covalent bonds with proteins through Schiff base or Michael addition, and silica gel binds to proteins through intermolecular hydrogen bonds. The binding stability of covalent bonds is higher than that of intermolecular hydrogen bonds, so the binding of SiO_2_-PDA to proteins is more stable.

### 2.5. Research on the Comparative Selectivity of CMC

The glucocorticoid receptor, which is the target of dexamethasone, is in the cytoplasm, while no target on the cell membrane has been identified [23,24]. Therefore, dexamethasone does not bind to membrane proteins and can be used as the negative compound for this experiment. Benzbromarone, a commercially available drug used for gout treatment, functions by inhibiting the activities of proteins situated on the cell membrane, such as URAT1 and OAT1 anion transporters, thereby reducing the reabsorption of UA. Consequently, benzbromarone can be used as a positive drug in CMC to screen for active compounds with anti-gout properties. The chemical structures of dexamethasone and benzbromarone are depicted in Appendix A.

This indicates that the SiO_2_-PDA/CMC columns showed higher column efficiency than the nondecorated group. The CMC columns show (Figure 5) a significant ability to distinguish between negative and positive compounds. Dexamethasone did not bind to membrane proteins and was not retained in the CMC column. Benzbromarone showed strong affinity on the CMC column, reaching a peak at 2.03 min on SiO_2_/CMC and at 8.96 min on SiO_2_-PDA/CMC. The retention time of Benzbromarone on SiO_2_-PDA/CMC columns exceeded 20 min, while the retention time on SiO_2_/CMC columns is only about 2 min. The SiO_2_-PDA/CMC columns bound more proteins, resulting in higher column efficiency. Benzbromarone exists in SiO_2_-PDA/CMC columns for a longer time than in SiO_2_-PDA/CMC, resulting in a longer molecular diffusion time and more severe effect in SiO_2_-PDA/CMC columns. This leads to a loss of column efficiency, which in turn affects the peak width, tailing, and fronting of the peak of benzbromarone in PDA columns. It has been proven that the number of proteins bound to SiO_2_-PDA is much higher than that to silica gel, which supports the existence of many proteins bound to benzbromarone on the surface of HK-2, such as URAT1 and OAT1. Therefore, this chromatographic column can serve as a new model for screening potential anti-gout drugs.

### 2.6. Screening Anti-Gout Compounds from the P2 Group of PAL

In this study, the SiO_2_-PDA/CMC column was used to screen potential anti-gout active compounds in P2. Figure 6 shows the chromatogram of leucoseptoside A in the SiO_2_/CMC and SiO_2_-PDA/CMC. The RT of leucoseptoside A on SiO_2_/CMC column was 0.79, and on SiO_2_-PDA/CMC was 1.85 min. This indicates that leucoseptoside A is retained on SiO_2_/CMC and SiO_2_-PDA/CMC columns and has a better retention effect in SiO_2_-PDA/CMC columns, which further proves that SiO_2_-PDA can bind more cell membrane proteins and have a better retention effect on compounds. A total of seven compounds were screened from the P2 group, all of which demonstrated retention in the CMC column, as shown in Table 1. The chemical structures of seven compounds can be observed in Appendix A. The compounds mentioned above exhibit potential as therapeutic agents for the treatment of gout. Taking leucoseptoside A as an example, all seven compounds exhibit tailing phenomenon in the CMC columns, resulting in a larger peak width. The later the peak time of a compound, the more severe the tailing phenomenon which is also related to the molecular diffusion of various compounds in the CMC columns.

### 2.7. UA Reabsorption in HK-2 Cells

In this study, the compounds screened using SiO_2_-PDA/CMC were subjected to further verification through a UA reabsorption experiment. The quantitative analysis of UA content was conducted using UHPLC-TQ-MS, with the utilization of hippuric acid as the internal standard. The precursor ion, product ion, Q1 (volt-age promotes the ionization of precursor ion), Q3 (voltage promotes the ionization of production), and collision energy (CE) of the two compounds were optimized (Appendix A). Appendix A displays the base peak intensity chromatograms of UA and hippuric acid.

HK-2 cells are commonly utilized in gout research due to their significance. URAT1, a highly expressed protein in HK-2 cells, belongs to the organic anion transport protein OATs and plays a crucial role in the reabsorption of UA [25,26]. In this experiment, UHPLC-TQ-MS was used to quantitatively detect the UA content of HK-2 reabsorption, indirectly reflecting the activities of URAT1. It is important to highlight that the presence of chloride ions can impact the reabsorption of UA; therefore, they should be excluded from the system [27,28].

In comparison to the control group (Figure 7), each experimental group demonstrates a significant reduction in the reabsorption of UA. This outcome indicates that the seven compounds effectively inhibit proteins associated with UA reabsorption, leading to a decrease in UA reabsorption. Consequently, the reliability of the CMC column in the experiment is further validated, and the occurrence of false positive results is eliminated.

### 2.8. SPR Analysis and Molecular Docking

Direct binding of acteoside to URAT1 was determined by SPR affinity analysis. The target immobilization level of URAT1 protein was 8500 response units (RU). As shown in Figure 8A, serial concentrations of ranging from about 3.125 μM to 50 μM were tested. The equilibrium dissociation constant (KD) for acteoside was calculated as 8.8 μM, indicating that acteoside was a potent compound binding with to URAT1.

Autodock was selected for molecular docking of acteoside and URAT1. Figure 8B shows the hydrogen bonds and active site. The binding energy of acteoside and URAT1 is −6.15 kcal/mol. This indicates a good combination of acteoside and URAT1.

## 3. Materials and Methods

### 3.1. Materials and Reagents

PAL was purchased from Hongjian Pharmacy (Changchun, China) and identified by Prof. Qing Huang (Jilin Academy of Traditional Chinese Medicine). HK-2 cell line was obtained from BeNa Culture Collection (Beijing, China). Dulbecco’s Modified Eagle Medium (DMEM) and fetal bovine serum (FBS) were supplied by Biological Industries Israel BeitHaemek Ltd. (Watertown, MA, USA). Ammonium acetate, UA, and hippuric acid were acquired from Sigma (St. Louis, MO, USA). Sodium gluconate, potassium gluconate, glutaraldehyde, dexamethasone, benzbromarone, and HCl·DA were purchased from Aladdin (Shanghai, China). Methanol, acetonitrile, and formic acid were HPLC-grade reagents obtained from Fisher Scientific (Lough borough, UK). Silica gel (5 μm, 200 Å, spherical) was supplied by Qingdao Makall Group (Qingdao, China). BCA protein assay kit and cell lysis buffer were purchased from Beyotime Biotechnology (Shanghai, China). Recombinant human protein URAT1 was purchased from Cloud-clone Corp (Wuhan, China). Solvents and all other chemicals not explicitly mentioned were of analytical grade and purchased from Beijing Chemical Works (Beijing, China).

The ultra-pure water was obtained from Milli-Q water purification system (Milford, MA, USA). Electric thermostatic drying oven was supplied by Yiheng Scientific Instrument Co., Ltd. (Shanghai, China). Allegra X-30R Centrifuge was obtained from Beckman (Brea, CA, USA). Ultrasonic cell crusher was supplied by Shanghai Zhengqiao Scientific Instruments Co., Ltd. (Shanghai, China). XL-30 Environment Scanning Electron Microscope (ESEM) was supplied by Philips (Amsterdam, The Netherlands). IFS 66 V/S FT-IR was obtained from Bruker (Karlsruhe, Germany). ASAP 2020 specific surface area analyzer was supplied by Mack Instruments (Atlanta, GA, USA).

### 3.2. Preparation and Component Identification of Samples Extract

According to our previous paper [20], 20 g of PAL in 200 mL of 75% ethanol (*v*/*v*) was extracted by Flash extractor. Then, solution was immersed for 1 h and refluxed for 2 h. The crude extract was concentrated to 2.5 g/mL and was eluted with water, 15%, 50%, and 70% ethanol (*v*/*v*) by an AB-8 macroporous resin column (20 mL, 1.6 cm × 20 cm), the volume of each eluent was 100 mL and flow rate was 40 mL/h. The eluates of 15%, 50%, and 70% ethanol was collected as three fractions (P1, P2, and P3), then concentrated and freeze dried three fractions for 24 h. The main components in P2 were analyzed by mass spectrometry, mainly phenylethanoid glucosides and flavonoids.

### 3.3. Preparation of SiO_2_-PDA

A mixture of 2 g of silica gel and 400 mL of HCl solution (1 mol/L) was subjected to heating reflux for 2 h. The resulting solid was subsequently washed three times with ultrapure water and subsequently dried in an electric thermostatic drying oven at a temperature of 120 °C for 16 h.

Different amounts of DA (0, 14, 28, 42, 56 mg), 50 mg of silica gel, and 500 mL of water were combined, and the pH was adjusted to 8.7 using ammonia water. The resulting mixture was subjected to gentle stirring using a magnetic stirrer for a duration of 2 h at room temperature. Subsequently, the mixture was washed three times with ultrapure water. The resulting mixture was then centrifuged at a speed of 5000 rpm for a duration of 10 min, and discard the supernatant and dried the soild t in a 60 °C electric thermostatic drying oven for a period of 24 h. The SiO_2_-PDA obtained from this process was placed in a dryer for future use.

### 3.4. Cell Culture, CMSP Preparation, and Characterization

The HK-2 cells were cultured in DMEM, supplemented with 10% (*v*/*v*) FBS and 1% penicillin-streptomycin solution at 37 °C in an incubator with 5% CO_2_. Subsequently, the HK-2 cells were digested using trypsin, counted using a cell counting plate, and subsequently washed three times with phosphate-buffered saline (PBS, 0.1 mol/L, pH = 7.2). Finally, the cells were heavily suspended in 2 mL PBS. The ultrasonic cell crusher was operated under the following conditions: a power of 400 W, ultrasound applied for 1 s, with an interval of 19 s for a total of 5 cycles. To prepare the HK-2 cells, they were first centrifuged at 1000 rpm for 10 min to remove the sediment. Subsequently, the supernatant was centrifuged at 12,000 rpm for 10 min, and the resulting precipitate was suspended in 5 mL of PBS. Finally, 40 mg of SiO_2_ or SiO_2_-PDA was added to facilitate the combination. The mixture was shaken under a vacuum for 5 min, followed by gentle stirring for a period of 30 min, and ultimately stored overnight. The entirety of the reaction process was maintained at a temperature of 4 °C. The final precipitate was washed with PBS three times to remove any unbound protein. Following this, the combined cell membrane protein was lysed using RIPA lysis buffer, and the total protein content was measured using the BCA kit.

In order to perform SEM testing on CMSP, it is necessary to dehydrate the CMSP, as follows: The CMSPs were subjected to stirring at room temperature in a 2.5% glutaraldehyde solution for a duration of 4 h. The resulting mixture was then sequentially dehydrated using ethanol solution of 30%, 50%, 70%, and 80% for 10 min each, and then dehydrated with anhydrous ethanol for 20 min. The filtered samples were subsequently dried and subjected to SEM analysis. Additionally, SiO_2_ and SiO_2_-PDA were prepared specifically for FT-IR and BET testing.

### 3.5. CMC Column Preparation

The CMSP was washed with 5 mL of PBS three times and subsequently filled with PBS to the column (10 mm × 2 mm I.D., 5 μm). The optimization of the packing flow rate was conducted using a linear gradient procedure, with the following parameters: 0–5 min, starting from 0.2 mL/min and gradually increasing to 1.0 mL/min; 5–6 min, maintained at 1.0 mL/min. Subsequently, the column was equilibrated at a flow rate of 0.2 mL/min for 30 min and a temperature of 37 °C until a stable column pressure and baseline were achieved. The CMC columns were stored in PBS at 4 °C.

### 3.6. CMC Analysis

The LTQ Orbitrap Elite MS (Thermo Scientific, San Jose, CA, USA) was used for CMC analysis. The capillary temperature was set at 320 °C, the tube lens voltage at 60 V, and the mass resolution at 60,000, and the maximum inject time was 100 ms. Prior to running the sample in this experiment, it was washed with 5 mmol/L ammonium acetate solution at a flow rate of 0.2 mL/min for a duration of 5 min. The resulting solution was then drained into a waste liquid tank to eliminate any presence of PBS and prevent its interference in the mass spectrum.

The freeze-dried powder of the P2 group of PAL was dissolved by using a small amount of DMSO (dimethyl sulfoxide), and subsequently diluted to a concentration of 100 μg/mL using methanol. The negative drug (dexamethasone, *m*/*z* = 392 Da) and the positive drug (benzbromarone, *m*/*z* = 424 Da) were both dissolved in DMSO and further diluted to a concentration of 2 mmol/L. The mobile phase consisted of a 5 mmol/L ammonium acetate solution, with a gradient of 0–25 min and a flow rate of 0.2 mL/min. The sample injection volume was 5 μL.

### 3.7. UA Reabsorption of HK-2 Cells

The culture medium used in this experiment does not contain chloride ions. Referring to the literature, the preparation of the culture medium is as follows [27]: 1.8 mmol/L KH_2_PO_4_, 10 mmol/L Na_2_HPO_4_, 140 mmol/L sodium gluconate, and 2.7 mmol/L potassium gluconate. The cells were incubated with an incubation medium containing either standards or benzbromarone (10 μg/mL) for 30 min at 37 °C. At the end of the incubation period, the medium was aspirated and the monolayers were rapidly washed twice with 1 mL of the incubation medium. Then, 0.4 mL solution of NaOH (20 g/mL) was added. Following this, 1 mL of methanol was added to induce protein precipitation. The mixture was centrifuged at 12,000 rpm for 10 min, and the resulting supernatant was dried using nitrogen. Subsequently, the samples were redissolved in 200 µL methanol solution containing 0.1 mol/L ammonia and 2 µmol/L hippuric acid, which served as the internal standard.

The quantitation of UA in the samples was performed by an ultra-high performance liquid chromatography coupled with triple quadrupole mass spectrometry (UHPLC-TQ-MS) system consisting of an Ultra-High Performance Liquid Chromatography LC-30A coupled with a triple quadrupole mass spectrometer LCMS-8060 by using an ESI source (Shimadzu Corp., Kyoto, Japan). An ACQUITY UHPLC BEH C18 column (50 mm × 2.1 mm, 1.7 μm, Waters, Milford, MA, USA) was used to separate the samples. The mobile phase consisted of A and B were acetonitrile and 0.1% (*v*/*v*) formic acid in water. The elution program was as follows: 0.0–4.0 min, 5–5% A; 4.0–8.0 min, 5–100% A. The sample injection volume was 5 μL. The flow rate and the column temperature were, respectively, set at 0.3 mL/min and 30 °C. The MS conditions for quantification analysis were optimized and finally performed in negative ion mode (ESI^−^) in multiple reaction monitoring (MRM) mode. The ESI source operation parameters were as follows: interface temperature 300 °C, DL temperature 250 °C, heat block temperature 400 °C, nebulizing gas 3.0 L/min, drying gas 10.0 L/min, and heating gas 10.0 L/min.

### 3.8. SPR Affinity Analysis

SPR assays were performed by Biacore T200 system (GE Healthcare, Gothrnburg, Sweden). Recombinant human protein URAT1 was diluted in running buffer (PBS-P^+^) to a concentration of 10 µg/mL and then immobilized on CM5 sensor chip (GE Healthcare, Gothrnburg, Sweden) in detection channel according to the manufacturer’s protocol. The detection temperature is set to 25 °C. The association and dissociation time were set to 60 s. The standard compound of acteoside was diluted at concentrations ranging from 3.125 μM to 50 μM. Analytes were injected at a flow rate of 50 μL/min. The affinity fitting was performed with Biacore T200 evaluation software 3.2 by global fitting using a steady-state affinity model to obtain the affinity constant.

### 3.9. Molecular Docking

Molecular docking simulation was processed by Autodock 4.2.6 software. The three-dimensional structure file of acteoside was downloaded from the Pubchem database. The human URAT1 structure file was downloaded from the AlphaFold protein structure database (ID: Q96S37). Acteoside was set as a ligand and saved as a pdbqt format file after dehydrogenation and polar hydrogenation. URAT1 was set as the receptor, and saved as a pdbqt file. Lamarckian genetic algorithm was selected as the docking algorithm. The number of docking conformations was set to 10, and the docking parameter file was saved during the process. After docking, the ligand conformation results and the receptor structure were retrieved, the appropriate docking conformation was selected and the complex structure file was exported.

## 4. Conclusions

The limited detection throughput and insufficient stability of chromatographic columns restrict the widespread application of CMC. Therefore, we optimized the CMC by using a PDA-modified carrier to enhance the protein loading capacity and protein binding strength of the CMC column. This method changes the binding mode between the carrier and the protein, resulting in a significant enhancement in protein loading. To verify and implement this novel CMC model, HK-2 cells were chosen as the optimal cells for evaluating potential compounds for treating gout, and the differences between SiO_2_-PDA/CMC and SiO_2_/CMC columns were compared. By combining mass spectrometry analysis, a high-throughput and highly stable HK-2/SiO_2_-PDA/CMC was established for the screening of potential anti-gout compounds in PAL. Seven potential anti-gout compounds were screened out. To verify these results, urate reabsorption experiments were conducted to eliminate false positive results, and SPR analysis and molecular docking demonstrate that acteoside and URAT1 can bind to each other. The optimized screening strategy significantly broadens the potential applications of CMC and facilitates its commercial viability.

## Figures and Tables

**Figure 1 ijms-25-01153-f001:**
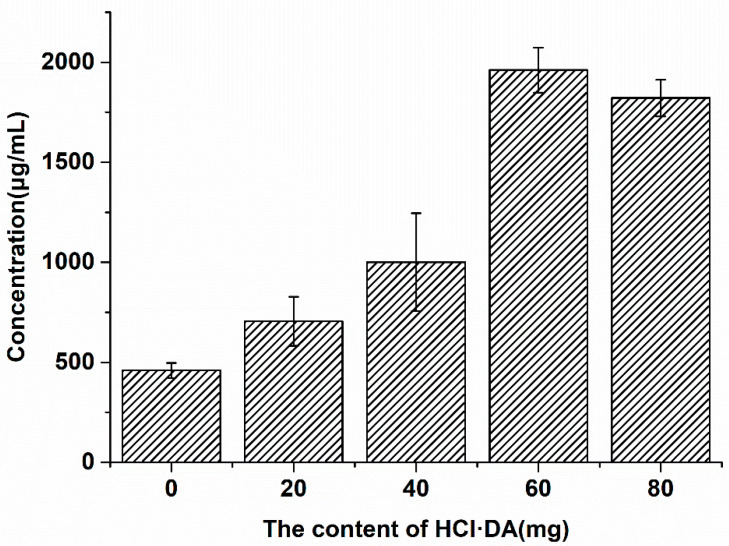
The saturated adsorption capacity of HK-2 cell membrane protein after combining HCl·DA with varying weights and 40 mg silica gel (*n* = 3).

**Figure 2 ijms-25-01153-f002:**
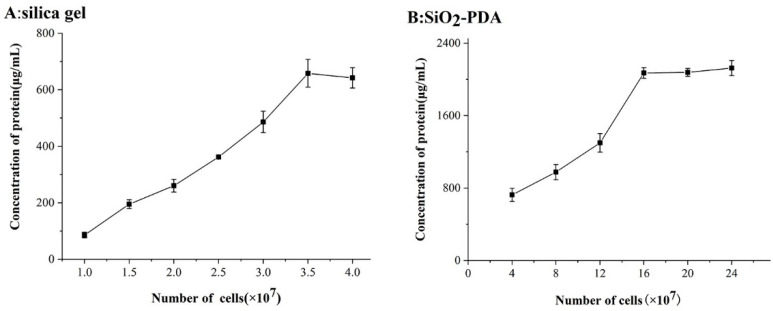
Immobilized protein quant on silica after incubation with multiple concentration of cell membrane suspension. (**A**) Content of membrane protein fixed on 40 mg silica gel; (**B**) content of membrane protein fixed on 40 mg SiO_2_-PDA. Data were expressed as mean ± SD (*n* = 3).

**Figure 3 ijms-25-01153-f003:**
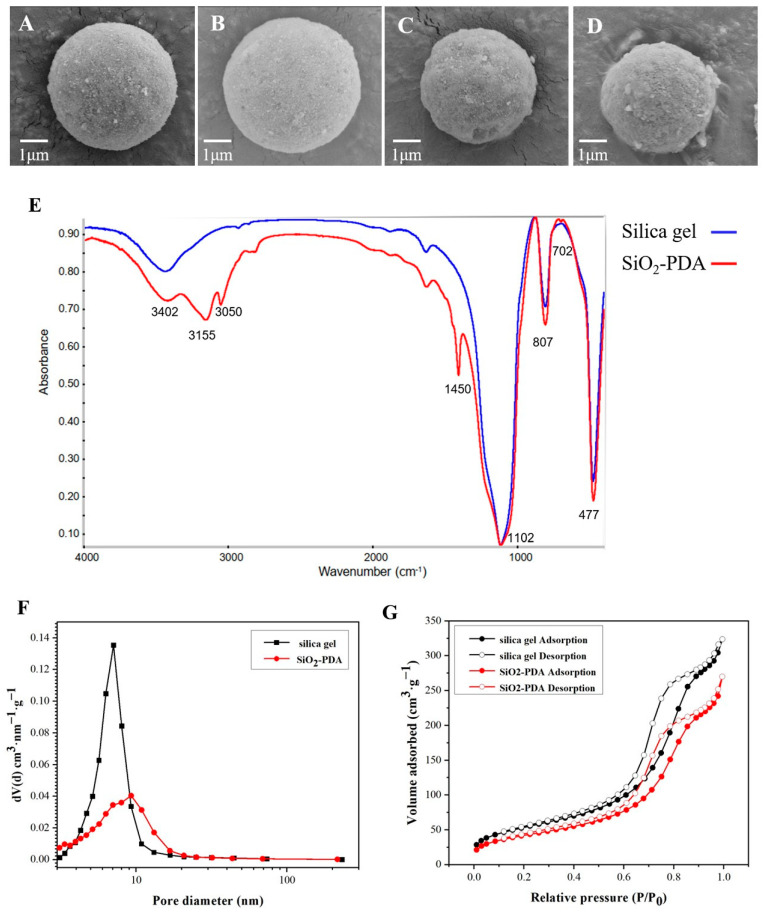
(**A**) SEM image of silica gel; (**B**) SEM image of SiO_2_-PDA; (**C**) SEM image of SiO_2_/CMSP; (**D**) SEM image of SiO_2_-PDA/CMSP; (**E**) FT-IR spectra of stationary phases; (**F**) the pore diameter of silica gel and SiO_2_-PDA; (**G**) adsorption and desorption isotherms of silica gel and SiO_2_-PDA.

**Figure 4 ijms-25-01153-f004:**
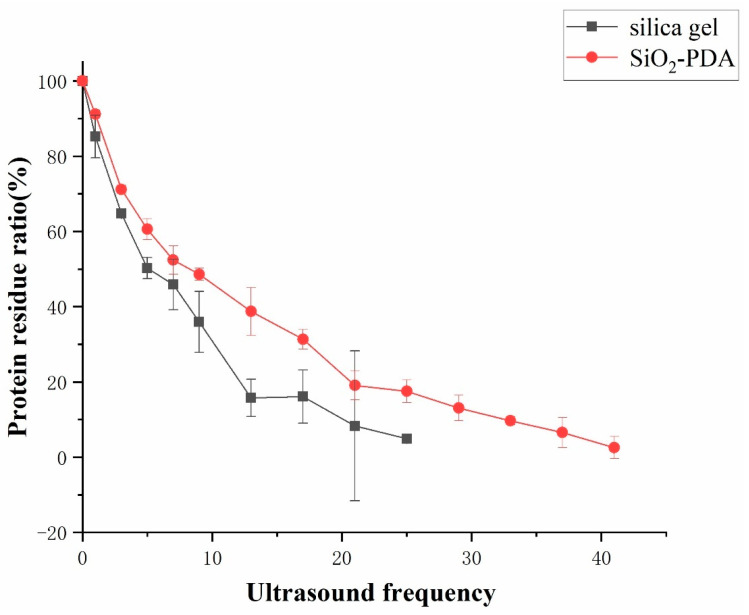
The residual ratio of protein on silica gel and SiO_2_-PDA after different ultrasound frequencies.

**Figure 5 ijms-25-01153-f005:**
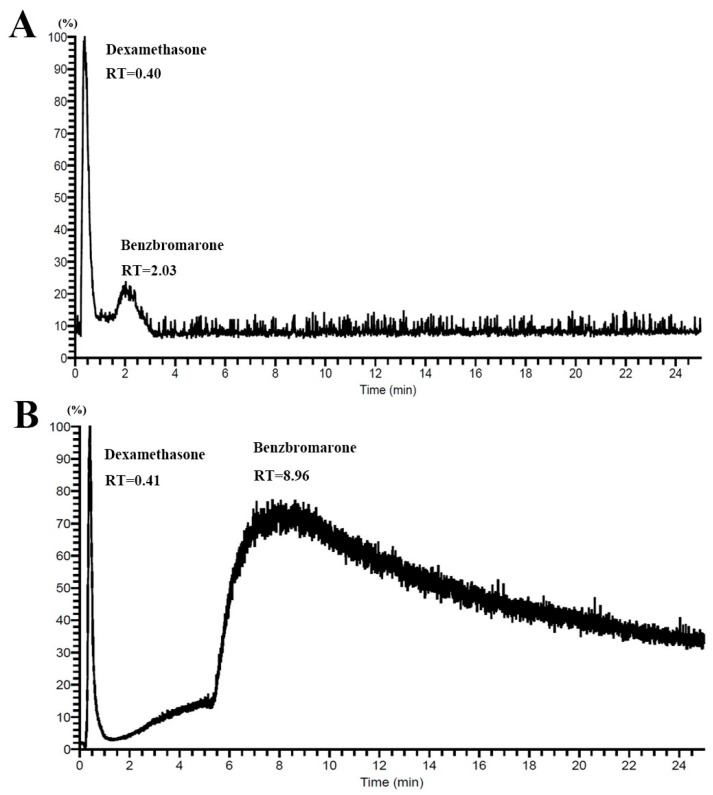
Chromatogram of dexamethasone and benzbromarone retained on the (**A**) SiO_2_/CMC and (**B**) SiO_2_-PDA/CMC column.

**Figure 6 ijms-25-01153-f006:**
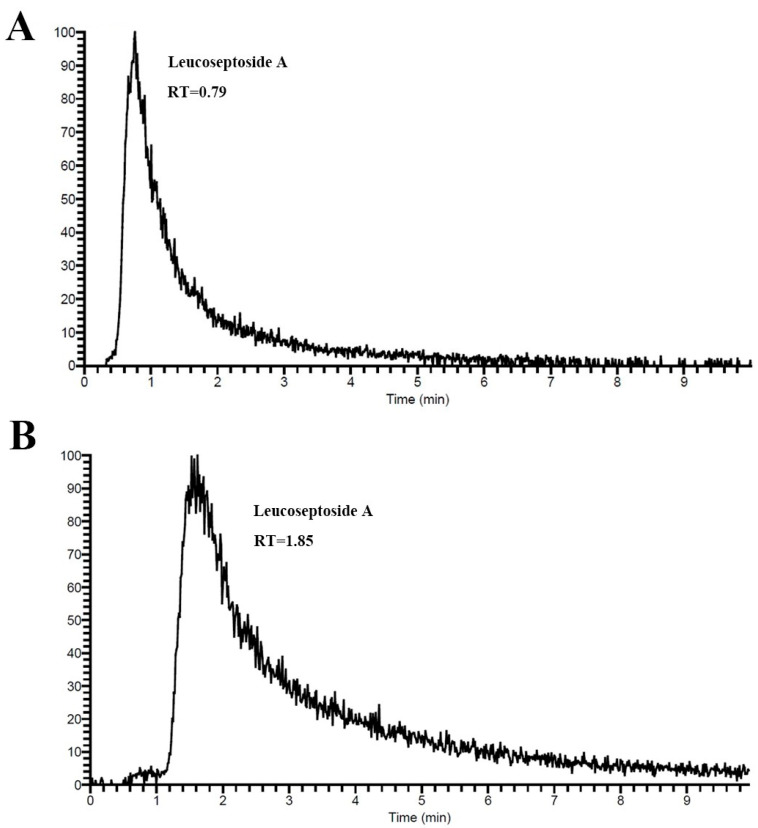
To screen compounds that may bind to HK-2 membrane receptor. (**A**) SiO_2_/CMC chromatograms of leucoseptoside A; (**B**) SiO_2_-PDA/CMC chromatograms of leucoseptoside A.

**Figure 7 ijms-25-01153-f007:**
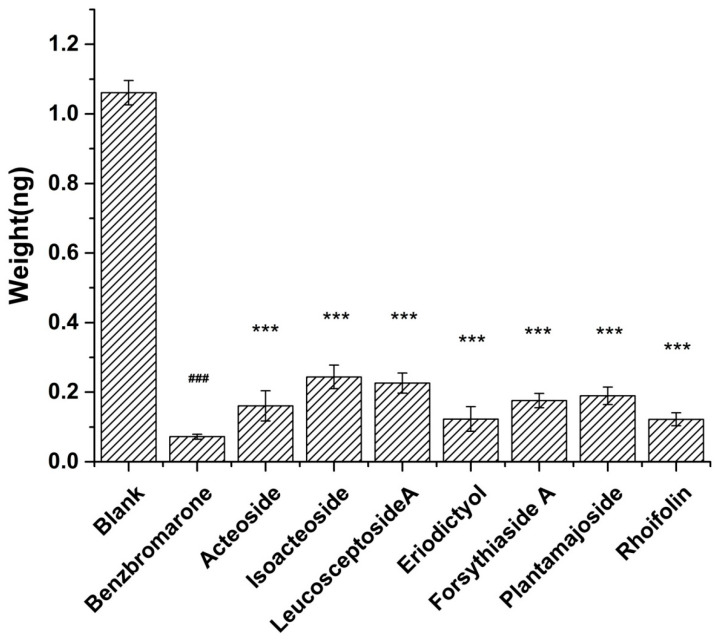
UA reabsorption in HK-2 cells. (*** *p* < 0.001 compared with model group; ### *p* < 0.001, compared with blank group).

**Figure 8 ijms-25-01153-f008:**
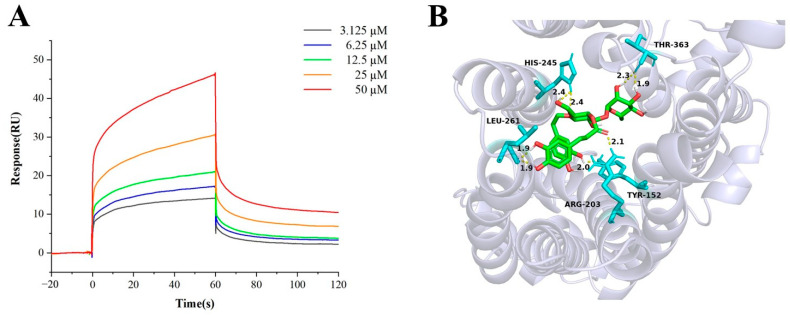
(**A**) Binding response curves of acteoside and URAT1 by SPR; (**B**) molecular modeling of interaction between acteoside and URAT1.

**Table 1 ijms-25-01153-t001:** Identification of the potential active compounds retained on SiO_2_/CMC and SiO_2_-PDA/CMC.

Compound	Calculated *m*/*z*	Observed *m*/*z*	RT on SiO_2_/CMC (min)	RT on SiO_2_-PDA/CMC (min)	Formula
acteoside	623.1976	623.2022	0.72	1.23	C_29_H_36_O_15_
isoacteoside	623.1976	623.2001	0.72	1.23	C_29_H_36_O_15_
forsythiaside A	623.1976	623.1992	0.72	1.23	C_29_H_36_O_15_
plantamajoside	639.1925	639.1956	0.76	1.85	C_29_H_36_O_16_
leucoseptoside A	637.2132	637.2151	0.79	2.50	C_30_H_38_O_15_
eriodictyol	287.0556	287.0560	1.05	2.75	C_15_H_12_O_6_
rhoifolin	577.1557	577.1572	1.35	2.95	C_27_H_30_O_14_

## Data Availability

Data are contained within the article and Appendix A.

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
