# Peer review of "Establishment of Polydopamine-Modified HK-2 Cell Membrane Chromatography and Screening of Active Components from Plantago asiatica L."

_ijms, 2024, doi:10.3390/ijms25021153_

Round 1

Reviewer 1 Report

Comments and Suggestions for Authors

1) The performance of the system should be compared with the other published studies. 

2) Please don’t use “Fig.” abbreviation instead of “Figure” in the text.

3) Line 119: Has DA been tested for binding to the SiO2 particle along with HCl?

4) DPA should be used instead of dopamine in the following sentence.

Line 122: “…. of dopamine, which covers the active site and reduces the active site per unit area”

5) P1, P2, and P3 abbreviations should be defined in the manuscript. 

6) The brand of devices that were used in the study should be presented in a separate part. The ultra-pure water system, centrifuge, oven, SEM, FT-IR, etc.

7) Please change the “r/min” unit to “rpm” in the manuscript.

8) Line 291: Is it “oven” or “stove”?

9) DMSO, SPR, CM5,  abbreviation should be defined at its first use. 

10) Please check the "dissolved" terms in the following sentence.

Line 345: “The cells were dissolved using a 0.4 mL solution of NaOH …”

11) m/z values for targeted compounds should be presented in the method part. 

12) The similarity rate of the parts 3.7 and 3.8 in the manuscript to other published studies is high. Please rewrite the related part accordingly. 

13) The measured compound should be written on the y-axis in Figure 2.

14) Average particle diameter together with their RSD should be presented. In addition, the BET surface area and pore size of both particles should also be added to the manuscript. 

15) “m/z” abbreviation should be added to Table S1. 

16) The unit of y-ordinate in Figure S4 should be added. 

17) The separation performance of a column is also related to the peak width, tailing, and fronting of the peak. The peak for benzbromarone is 8.96 min is not indicating a proper separation in Figure 4. Please discuss it in the manuscript. 

18) The performance of SiO2 and SiO2-PDA should be compared in the experiments.

19) The unite of the y-axis in all figures should be presented. 

20) Images and subtitles of Figure 5 should be placed on the same page.

21) The order of Table 1 should be presented according to retention times. 

Reviewer 2 Report

Comments and Suggestions for Authors

The current paper presents a HK-2/SiO2-PDA/CMC method for the screening of seven potential anti-gout compounds derived from Plantago asiatica L, based on their binding affinity towards the membrane protein of HK-2 cells.

Polydopamine is one of the simplest and most versatile approaches to functionalizing material surfaces. Therefore, the authors of this study, utilize this material to develop a new CMC column in order to increase protein loading and improve detection flux. This CMC column uses polydopamine (PDA) coated silica gel formed by self-polymerization of dopamine (DA) as the carrier of CMC column filler, and the human kidney 2 (HK-2) cell membrane proteins were immobilized on silica gel with PDA coating (SiO2-PDA). The ratio of PDA and silica gel and the cell dosage were optimized. The obtained cell membrane stationary phase was characterized by SEM, FT-IR and the effect of suction filtration was investigated. The suitability of the new column to screen anti-gout compounds was demonstrated using dexamethasone as negative control (RT = 0.41 min) and benzbromarone (an anti-gout drug inhibitor of URAT1 and OAT1) as positive control (RT = 8.96 min). The SiO2-PDA column was further used to screen potential anti-gout active compounds from an extract of Plantago asiatica L obtained, characterized and presented in a previous paper of the same authors. The seven compounds effectively inhibit proteins associated with UA reabsorption, leading to a decrease in UA reabsorption. The results were verified by urate reabsorption experiments conducted to eliminate false positive results, surface plasmon resonance analysis and molecular docking demonstrate that acteoside and URAT1 can bind to each other.

The novelty of the current paper consists in the development of a HK-2/SiO2-PDA/CMC model for screening potential therapeutic drugs for gout. The study is well designed, the method and reagents are described with sufficient details and the conclusions are justified. However, the text can be improved by a more detailed expression of certain mechanisms or properties. Some abbreviations are not explained in the text and a typo was noticed.

The study is well designed, the method and reagents are described with sufficient details and the conclusions are justified. However, some minor modifications are needed:

Line 212: Fig. 4 should be replace by Fig.5.

Reviewer 3 Report

Comments and Suggestions for Authors

Title: Establishment of PDA-modified HK-2 cell membrane chromatography and screening of active components from Plantago asiatica L.

Recommendation: Minor revisions needed as noted.

This manuscript presents a study on the development of a novel Cell Membrane Chromatography (CMC) column utilizing polydopamine-coated silica gel for screening potential therapeutic drugs for gout. Here are some questions and suggestions for revision:

1.      Could you provide more details on how covalent bonding with proteins is introduced in the novel CMC column? What advantages does this confer?

2.      Beyond greater affinity for membrane proteins, are there other advantages of SiO2-PDA over silica gel in terms of stability or reproducibility?

3.      Can you quantify or provide more details on the improvement in detection efficiency achieved using SiO2-PDA compared to silica gel?

4.      Beyond the development of the novel CMC filler, can you elaborate on how this technology may practically impact drug discovery for gout treatment?

5.      You mentioned that the binding between SiO2 and protein is through intermolecular hydrogen bonding, while PDA and protein binding involve covalent bonds. Could you elaborate on the differences in stability between these two binding mechanisms?

6.      How was the optimal ratio of dopamine polymerization to the SiO2 surface determined, and what criteria were considered in defining the optimal ratio?

7.      Can you provide more insights into why an excessive concentration of dopamine is disadvantageous and how it affects the binding of membrane proteins?

8.      How was the optimal quantity of cells determined for the preparation of the CMC column? Could you elaborate on the significance of minimizing cell dosage and its impact on the efficiency and performance of the CMC column?

9.      How was the CMC column's ability to differentiate between negative (dexamethasone) and positive (benzbromarone) compounds assessed? What insights does the chromatogram provide regarding the selectivity of the CMC column?

10.  How does the UA reabsorption experiment validate the reliability of the CMC column in screening potential anti-gout compounds? Can you provide more details on the mechanism through which the identified compounds inhibit proteins associated with UA reabsorption?

Comments on the Quality of English Language

Minor changes are needed.

Round 2

Reviewer 1 Report

Comments and Suggestions for Authors

In the revised version of the manuscript required corrections have been made in the article. On the other hand, there are some points that should be corrected before publication. Related points are listed below.

1)      In the authors' response it was stated that only DA binds to the surface without HCl. If the adsorption of DA is not observed together with HCl, please use only DA instead of DA.HCl in the corresponding sentence and figure (Figure 1). Please check the unit of y ordinate in Figure 1.

2)      Line 202: Ultrasonication time should be checked in the following sentence (1 second or 1 minute).  

“The conditions for ultrasound are: a power of 400W, ultrasound applied for 1 s, with an interval of 19 s.”

3)      Please revise the following sentence.

Line 205: “. In comparison, after ultrasound 41 times, the proteins on SiO2-PDA were completely detached."

4)      Retention time and area stabilities of the column for the analytes should be mentioned.

5)      If it is possible, the performance of the developed column for the target analytes should be compared with other columns used for the analysis of these analytes in the literature.
